# Plant Growth Promoting and Colonization of Endophytic *Streptomyces albus* CINv1 against Strawberry Anthracnose

**Waraporn Pupakdeepan** [1,2], **Natthida Termsung** [1], **On-Uma Ruangwong** [1] **and Kaewalin Kunasakdakul** [1,*]

1 Department of Entomology and Plant Pathology, Faculty of Agriculture, Chiang Mai University, Chiang Mai 50200, Thailand; waraporn_pupak@cmu.ac.th (W.P.); natthida.ts@gmail.com (N.T.); on-uma.r@cmu.ac.th (O.-U.R.)

2 Faculty of Agriculture, Uttaradit Rajabhat University, Uttaradit 50300, Thailand

* Correspondence: kaewalin.k3@gmail.com; Tel.: +66-94941-6223

**Abstract:** Strawberry anthracnose is a serious disease, and fungicides are currently widely used by farmers. Thus, biological control is a good alternative. This study aims to identify the species of endophytic *Streptomyces* CINv1 that was previously isolated from *Cinnamomum verum* J. Presl. and to evaluate its properties as a biocontrol agent, plant growth promoter, and plant colonizing endophyte. This strain was identified by analysis of its 16S rRNA gene sequences, and the result shows 100% similarity to *Streptomyces albus* CINv1. The CINv1 strain displayed high resistance (81.83%) against *Colletotrichum* sp. isolate CA0110, as tested by the dual culture technique. Additionally, inhibited pathogen growth on IMA-2 agar was observed under a compound microscope. The results demonstrated swelling, bulbousness, and cytoplasmic aggregation of abnormal hyphal, which were confirmed by SEM as well. Furthermore, the functional media used to evaluate plant growth-promoting properties, including nitrogen fixation, phosphate solubilization, and siderophore production, yielded positive results. Analyses of plant hormones by HPLC found their ability to produce indole-3-acetic acid (IAA). Thus, a biological control trial in greenhouse conditions was conducted by spraying a spore suspension of the strain onto strawberry seedlings once a week, which showed a significant reduction in disease severity. After the seventh spraying, the assessment of the number of leaves and canopy height of the seedling showed significant promotion. In addition, the CINv1 strain established a mutualistic interaction with the plant cells through colonization inter-and intracellularly in strawberry roots, leaves, and petioles. Moreover, using LC-MS/MS to analyze the secondary metabolites of this strain, various groups of compounds were found that could potentially benefit pharmaceutical and agricultural uses.

**Keywords:** colonization; endophytic streptomyces; plant disease control; plant growth promotion; secondary metabolites

## 1. Introduction

Strawberry (*Fragaria* × *ananassa* Duch.) is one of the most popular horticultural crops in many countries, with an annual production volume of about 6.1 million tons worldwide [1]. Chiang Mai province in northern Thailand is the center of strawberry production in the country, with an economic value of more than THB 200 million per year from 2005 to 2010 [2]. Strawberry cv. Pharachatan 80, the most popular variety, was developed by The Royal Project Foundation for commercial purposes [3]. Strawberry anthracnose is a common disease of strawberry production in the highland region of Chiang Mai, Thailand. This primary disease of strawberry fruit is caused by *Colletotrichum* spp. [4]. Three species of Colletotrichum, *C. fragariae* (Books), *C. gloeosporoides* (Penz), and *C. acutatum* (Simmonds), are responsible for strawberry fruit anthracnose disease [2,5]. In Thailand, it has been shown to be caused by *C. acutatum* [2,6]. However, fungal infections can occur in both the preharvest and postharvest stages, causing severe damage to fruit and economic losses [7].

In recent decades, however, several new fungal species have been discovered as causal agents of strawberry anthracnose, such as *C. miaoliense*, *C. karstii*, *C. siamense*, *C. fructicola*, and *C. boninense* [8]. Fungicides are currently widely used by farmers in the cultivation of strawberries. These toxic substances pose a risk to humans and the environment when used improperly and excessively [2,9]. In addition, the pathogen may be resistant to fungicides and phytotoxic agents [10]. In recent years, interest in agriculture production has increased due to concerns about protecting the upland environment and human health. Strawberry pest control using environmentally friendly approaches such as biological control agents and integrated pest management is more acceptable [11]. To address these issues, scientists have worked to develop environmentally friendly biological control agents for soilborne plant diseases [10]. In Thailand, antagonistic bacteria have been shown to be effective biological control agents for strawberry anthracnose. Thus, six strains of *Streptomyces* sp. were evaluated for their antifungal activity against *C. gloeosporioides* isolate Cg_MCL8 [12] and the Antagonistic bacteria *Bacillus subtilis* isolate K27 [2]. Nevertheless, they are very few and deserve further study.

Endophytes refer to microbial communities, including bacteria, archaea, and fungi, that live in plant tissue without causing visible disease symptoms or negative effects on the host [13]. Endophyte bacteria help the plant absorb nutrients, improve stress tolerance, provide disease resistance, promote plant growth, and are beneficial for agriculture sustainability [14–16]. Successful colonization in the host plant cell is a key factor in beneficial plant-microbe interactions, leading to various plant growth-promoting mechanisms. The inlet or colonization of the host plant by endophytic bacteria is a complicated phenomenon and involves a set of events [13].

Actinomycetes are gram-positive bacteria that are mostly aerobic, but some of them can also grow anaerobically. Some actinomycete species produce external spores, while others have branching filaments and exhibit mycelial growth. Actinomycetes are considered unique among rhizosphere microorganisms [17]. They are regarded as special in promoting plant growth because they display a variety of advantageous traits [18] and the ability to colonize plant roots. *Streptomyces* are important groups of soil bacteria from the Actinomyces family. *Streptomyces* is one of the major known sources of bioactive metabolites [19]. Bioactive substances include the production of secondary metabolites in the form of antibiotics [19,20], and antimicrobials are used in agriculture and medicine [21]. Seventy-five percent of natural antibiotics come from the genus *Streptomyces*, and about two-thirds of them have been isolated from Actinomycetes [22]. *Streptomyces* commonly inhabit soil [23], and rhizospheres and are known to antagonize pathogens by producing a variety of bioactive secondary metabolites [18,24] such as antibiotic [25,26] and antifungal [27]. In addition, they are widely known to synthesize extracellular proteins and produce various lytic enzymes [27]. In addition, they can stimulate productivity and provide productivity-stimulated, Plant-mediated management of plant diseases and pests, as well as plant growth promotion [28].

Plant growth-promoting *Streptomyces* (PGPS) are effective colonizers of both the rhizosphere and rhizoplane. They could potentially be endophytes that colonize the host plant's tissue [17]. This ability may be due to features such as controlled gene expression by quorum sensing, amino acid synthesis, cellulase, lipase, and β-1,3-glucanase production, antibiotics, phytohormones, and siderophores. Plant rhizosphere exudates stimulate *Streptomyces*' chemotactic movement [18]. The use of PGPs as biological control agents or biofertilizers in agriculture has led to new research on the potential applications of *Streptomyces* [18,29]. However, little information is available on the use of endophytic *Streptomyces* as fungal antagonists and growth promoters of the strawberry plant. Thus, the aim of this study is to identify endophytic *Streptomyces* CINv1 and investigate its potential biological control of strawberry anthracnose disease and the mechanisms of plant growth promotion and inter-intracellular colonization. This study contributes to the development of biological control agents, bio-based products, and fungicide reduction for these strains for sustainable agriculture in the future.

## 2. Materials and Methods

### 2.1. Confirmation of Strawberry Anthracnose Fungal Pathogen

Isolation, Pathogenicity Tests, and Identification of Colletotrichum sp.

The phytopathogenic fungus *Colletotrichum* sp. was isolated from infected strawberry cv. Prarachatan No.80. In brief, strawberry fruits were sampled in a local strawberry field in Mae Rim district, Chiang Mai province, Thailand. Each infected fruit was washed with sterile distilled water, cut into smaller pieces, and dipped in 10% Clorox for 3 min. It was rinsed with sterile distilled water at least three times, transferred onto a Petri dish containing potato dextrose agar (PDA), and incubated at 25 °C for 7 days. After this, all fungi were subcultured to obtain a pure culture, and the isolated phytopathogen was pre-identified based on morphological characteristics including colony morphology, the appressorium, conidiogenous cells, and conidia [30,31]. Then, each strain was tested for pathogenicity in order to select a virulent isolate according to a slight modification of the method described by Wang et al. and Cacciola et al. [32,33]. Briefly, ten detached strawberry fruits and leaves were wounded with a sterile needle and inoculated with a plug of *Colletotrichum* sp. Fruits and leaves were kept in a moist plastic box and incubated at 25 °C for 7 days. Control samples were inoculated with a PDA agar plug. The test was repeated three times. The identification of isolate CA0110 was confirmed by DNA sequencing. The aerial mycelium of *Colletotrichum* sp. isolate CA0110 was scraped from the PDA surface and pulverized with a mortar and pestle to achieve a mycelium pulp. The extraction of genomic DNA and polymerase chain reaction (PCR) were performed following a protocol described by Tanapichatsakul et al. [34]. The internal transcribed spacer (ITS) region was amplified using ITS5/ITS4 primers. Meanwhile, an intron region was amplified using glyceraldehyde-3-phosphate dehydrogenase (GAPDH), including GDF1 and GDR1 primers [35].

### 2.2. Characteristic Identification of Endophytic Streptomyces CINv1

2.2.1. Re-Isolation and Morphological Study of Endophytic Streptomyces CINv1

Endophytic *Streptomyces* CINv1 strain, which was previously isolated from the Thai mediational plant (*Cinnamomum verum* J. Presl.), was obtained from the Virology Laboratory, Department of Plant Pathology and Entomology, Faculty of Agriculture, Chiang Mai University. The strain was subcultured on IMA-2 medium for 7 days. Then, the hyphal morphology data, including colony color and the arrangement of the spore chain, were examined using a Scanning Electron Microscope (SEM), and micrographs were produced using the FEI-Quanta 450 at the Synchrotron Light Research Institute, Nakhon Ratchasima, Thailand [36].

2.2.2. Molecular Identification of Endophytic Streptomyces CINv1

Molecular identification of *Streptomyces* CINv1 was performed using the 16S rRNA gene for amplification by PCR. Firstly, genomic DNA was extracted from fresh cells from 1 plate on IMA-2 at 30 °C for 7 days; mycelium was scraped and ground in liquid nitrogen to a fine powder. The extraction of genomic DNA was carried out step by step using a previously published Gram-positive bacteria genomic DNA purification protocol (Thermo Fisher Scientific, Massachusetts, WA, USA), with slight modifications [37]. The quality and quantity of extracted DNA were measured by agarose gel electrophoresis and a NanoDrop machine (Thermo Fisher Scientific, Massachusetts, WA, USA). Polymerase chain reaction (PCR), conditioning, and visualization of products were performed according to the method described by Chaiharn et al. [38]. PCR amplification was performed using KOD OneTM PCR Mastermix Blue-Master Mixes (Toyobo, Tokyo, Japan) in thermal cycle PCR (Biometra, Gottingen, Germany). For molecular identification, DNA was amplified using a 16S rRNA gene primer set (forward (27F) and reverse (1525R); the materials and cycles of PCR amplification are described by Chaiharn et al. [38], adopted with slight modifications. In this study, we used 1.5 μL of 10 μM primer and 1 μL of 50 ng chromosomal DNA, and the final volume used was 50 μL. The PCR amplification was performed using a thermal

cycler (Veriti Applied Biosystems, Waltham, MA, USA) according to the following protocol: 94 °C for 4 min as a primary denaturation step, 30 cycles of 94 °C for 1 min, 55 °C for 1 min, 72 °C for 2 min, and final extension at 72 °C for 10 min. The PCR products were visualized using gel electrophoresis on 1.7% agarose (Qiagen, Hilden, Germany) at 120 V for 45 min and compared with 1.5 kb DNA ladder (Fermentas, Leon-Rot, Germany). After PCR amplification, the PCR product was sequenced (Macrogen, Seoul, Republic of Korea), and phylogenetic trees were constructed using the program MEGA [39].

SeqMan 5.00 software was utilized to obtain the contig sequence after thoroughly examining the sequences for any inconsistencies. The contig sequence was subjected to the BLASTn search in NCBI BLAST search to find similar hits and assess the quality of the sequence. Based on BLAST results, those from other studies, and the GenBank database (Considering the high query coverage and high identity), relevant species and strains were chosen. The related sequences were downloaded from NCBI GenBank. The dataset was prepared using MEGA V.7 [40]. The dataset was aligned using MAFFT version 7 [41]. Manual adjustments were made using MEGA V.7 to incorporate the necessary changes. Subsequently, phylogenetic trees were reconstructed using the MEGA neighbor-joining method along with bootstrap resampling, approaching 1000 replications to evaluate the confidence values of the nodes.

### 2.2.3. Inhibition Efficiency of Streptomyces CINv1 on the Radial Growth of Colletotrichum sp. Isolate CA0110

The antifungal activity of *Streptomyces* CINv1 against *Colletotrichum* sp. isolate CA0110 was tested using the dual culture method [42]. Briefly, endophytic actinobacterium isolates CINv1 were cultured on two sides, at a distance of 1.5 cm from the edge of a Petri dish containing IMA-2 medium, and incubated at 30 °C for 5 days [43]. After this, the mycelial disc (0.5 cm diameter) of the fungal pathogen was placed in the middle of the plate. Sterile distilled water was used as a control treatment. The antagonistic activity of the CINv1 strain was expressed as the mean of ten replications. The radius of the fungal colony in each plate was measured after 7 days of incubation. The percent inhibition of radial growth (PIRG) was calculated with Equation (1) [44]:

$$PIRG = (R1 - R2/R1) \times 100 \tag{1}$$

R1 = Radial mycelium growth of the untreated control

R2 = Radial mycelium growth of treated treatment

### 2.2.4. Scanning Electron Microscopy (SEM)

Petri dishes (IMA-2 medium) with and without *Streptomyces* CINv1 were inoculated with one cock of *Colletotrichum* sp. isolate CA0110 and then incubated for 7 days. Scanning Electron Microscope (SEM) specimen preparation: the samples were cut at the end of the fungal hyphal and transferred into the 1.5 mL sterile Eppendorf tubes, and 2.5% glutaraldehyde in 0.1 phosphate buffer pH 7.3 was added at 4 °C for 3 h. Then, the specimen was washed with phosphate buffer three times for 10 min, and the samples were then fixed in 2% osmium and dehydrated in an alcohol series at 50, 70, 85, 90, and 100%. The samples were dried in a Critical Point Dryer (CPD) and coated via SEM [36,45]. Micrographs were produced using the JSM-IT300 SEM at the Central Science Laboratory, Chiang Mai University, Thailand.

### *2.3. Plant Growth Promoting Activities*

### 2.3.1. Phosphate and Potassium Solubilization Ability

Pikovskaya's modified medium was used to determine the phosphate solubilization ability for the qualitative assay. Meanwhile, solubilization of potassium was determined in Aleksandrov's modified agar medium containing bromothymol blue for the qualitative assay [46]. Freshly grown bacterial cultures were inoculated on the agar plate and incubated

at 30 °C for 7 days. Phosphate solubilization and potassium solubilization were determined by measuring the clear halo around the colony [47].

### 2.3.2. Nitrogen Fixation

The nitrogen-fixing capability was determined by using Burk's medium, which allows the growth of only nitrogen-fixing bacteria [48]. Freshly grown *Streptomyces* CINv1 cultures were inoculated on an agar plate and incubated at 30 °C. The positive result of nitrogen fixation activity can be determined by the presence of the study colony after 5 days of inoculation.

### 2.3.3. Siderophore Production

Siderophore production was assayed using Chrome Azurol's Blue agar (CAS) for the qualitative assay. *Streptomyces* cultures were spotted on CAS plates and incubated for 5 days at 30 °C in dark conditions. The results of *Streptomyces* CINv1 culture, indicating a yellow-to-orange-colored zone around colonies, indicated the production of siderophores [49].

### 2.3.4. Auxin Quantification

Endophytic *Streptomyces* CINv1 was cultivated on IMA-2 agar and incubated at 30 °C for 7 days. The agar was corked into the international streptomyces project-2 medium (ISP-2 medium). L-Tryptophan (2 mg/mL) was added to the ISP-2 medium and shaken with a rotating shaker (Digital Orbital Shaker; SCILOGEX SK-0330-Pro) at 150 rpm at 30 °C for 7 days. The culture filtrate of CINv1 was extracted with ethyl acetate as a solvent and dried with a rotary evaporator at 40 °C (Hei-VAP Precision). Crude extracts were dissolved in methanol for the determination of indole-3-acetic acid (IAA) by HPLC [50].

### *2.4. Hydrolytic Enzyme Production*

#### 2.4.1. Detection of Protease Synthesis

Protease activity was screened on 7% skim milk agar containing 5 g pancreatic digest of casein (peptone C), 2.5 g yeast extract, 1 g glucose, 0.7 g skim milk solution, and 15 g of agar (per liter) (pH 7). *Streptomyces* CINv1 plugs (6 mm) were placed on the medium and incubated for 5 days at 30 °C. Proteolytic activity was identified by the clear zone around the plugs [51].

#### 2.4.2. Detection of Cellulase Synthesis

Tested *Streptomyces* CINv1 were grown on 2.5% CMC agar for 5 days at 30 °C, flooded with 1% Congo red solution, and washed with distilled water. The clear zone around the colony was observed and measured [51].

#### 2.4.3. Detection of Chitinase Synthesis

Screening for chitinase production was performed by agar plate assay on a colloidal chitin medium containing 1.5 g colloidal chitin, 0.5 g yeast extract, 1 g $(NH_4)_2$ $SO_4$, 0.3 g $MgSO_4$ $6H_2O$, 1.36 g $KH_2PO_4$, 15 g agar, and 1000 mL distilled water. The plates were incubated for 5 days at 30 °C, and a clear zone around the colony presented chitinase activity [51].

#### 2.4.4. Detection of Amylase Synthesis

Amylase activity was screened on starch agar containing peptone (5.0 g/L), beef extract (3.0 g/L), soluble starch (2.0 g/L), and agar (16 g/L). Isolate suspensions were inoculated onto starch agar media and incubated for 7 days at 30 °C. Dishes were flooded with iodine solution, and the diameters of positive results were measured by the non-blue surrounding growth [51].

*2.5. Plant Growth-Promoting Activity in Tissue Culture Conditions*

2.5.1. Preparation of Strawberry Plantlets

Runners of strawberry plant cultivar Prarachatan No.80 were used as material to produce plantlets with the meristem culture according to the method of Naing et al. [52], which was slightly modified. The 2-cm-long collected runners were thoroughly washed under tap water for 15–30 min. The aseptic sterilization of the runner tip was achieved by dipping the tips in sterile distilled water containing 1% sodium hypochlorite (NaOCl) and a drop of Tween 20 for 10 min; then, they were rinsed with sterile distilled water at least three times. This material was used as an explant for meristem culture and was then transferred to a new free-hormone medium. The explants were incubated in a culture room at 25 °C under a photoperiod of 16 h.

2.5.2. In Vitro Plant Growth Promotion Traits

The 60-day-old strawberry plantlets with 3 leaves were cut and transferred to 20 mL of new free-hormone tube medium in a tube and incubated in the tissue culture room. After 10 days, the plantlets were inoculated with 10 μL/tube of spore suspension at $10^6$ cfu/mL of *Streptomyces* CINv1 by dipping the plantlets and incubating them in the tissue culture room for 30 days. Each treatment consisted of thirty explants. Sterile distilled water was used as a control treatment. After 30 days, the parameters of plants were measured, including the number of leaves, number of roots, number of stems, canopy height (cm), root length (cm), and total fresh weight (g) [53].

2.5.3. Plant Growth Promotion Traits and Biological Control in Greenhouse Conditions

The 60-day-old strawberry plantlets were grown in a pot that contained mixed soil and perlite (3:1), cultivated in an evaporative cooling system, and inoculated with 10 mL/pot of spore suspension at $10^6$ cfu/mL of *Streptomyces* CINv1 by spraying every week, 7 times. Sterile distilled water was used as a control treatment. Each treatment comprised eight replications per treatment. The parameters of plants were measured, including the number of leaves, number of runners, and plant height (cm) [53,54]. Subsequently, the strawberry plant was inoculated with $10^4$ cfu/mL of spore suspension of *Colletotrichum* sp. isolate CA0110, and after inoculation, the disease incidence and severity were measured. The percentage of incidence of the disease was determined according to Equation (2), and 7 days after inoculation of the pathogen, the disease severity score was determined as follows: score 0 = no symptom, 1 = very slight, 2 = moderate, and 3 = severe and dead plant [54].

$$\text{\% Incidence} = (\text{Infected plants/Total plants}) \times 100\% \qquad (2)$$

*2.6. The Study of Colonization of Streptomyces CINv1*

Three types of strawberry plantlet tissue (leaf, petiole, and root) were cut to a size of $1 \times 1$ cm and transferred to a Streptomyces CINv1 spore suspension ($10^6$ cfu/mL) for 3–5 min and loaded into a water agar plate (WA). Then, the sample plates were incubated at 25 °C for further use.

Scanning Electron Microscope (SEM) specimen preparation: the samples were transferred into the 1.5 mL sterile microcentrifuge tube, and 2.5% glutaraldehyde in 0.1 phosphate buffer pH 7.3 was added at 4 °C for 3 h. Then, the samples were washed with phosphate buffer three times for 10 min, and the samples were then fixed in 2% osmium and dehydrated in an alcohol series at 50, 70, 85, 90, and 100%. The samples were dried in a Critical Point Dryer (CPD) and coated via SEM [36,45]. Micrographs were produced using the JSM-IT300 SEM at the Central Science Laboratory, Chiang Mai University, Thailand.

The leaves, petiole, and roots of strawberry plantlets were trimmed to cubes of 1.0–1.5 mm size and fixed by immersion in 2–3% (*v*/*v*) glutaraldehyde in 0.1 M sodium cacodylate buffer, pH 7.2, at 4 °C overnight. The samples were post-fixed in 2% osmium tetroxide-buffered solution and embedded in epoxy resin. Subsequently, the samples were cross-sectioned (0.1 μm) with ultra-microtome and stained with a saturated solution of

uranyl acetate and lead citrate. Micrographs were produced using the FE-TEM-Thermo Scientific TALOS F200X at the Synchrotron Light Research Institute Nakhon Ratchasima, Thailand [36,45].

### 2.7. Screening of Secondary Metabolites Produced by Streptomyces CINv1

The chemical compounds produced by *Streptomyces* CINv1 were identified by Liquid Chromatography/Tandem Mass Spectrometry (LC-MS/MS): 6200 series TOF/6500 series Q-TOR B.08 version 8058.0 mass spectrometer operating in electrospray ionization (ESI) negative mode and hyphenated with an Agilent 1290 ultra-high performance liquid chromatography system. The high-purity nitrogen gas for the mass spectrometer was set following the method of Awla et al. [55] at 40 psi for the source gas, 40 psi for the heating gas, and high for collision gas with a source temperature of 500 °C. The setting for electrospray ionization voltage was set at 4500 kV. The collision energy to attain fragmentation was set at 35 eV with a spread of $\pm15$ eV. The MS/MS scan mass range was set from 50–1000 $m/z$, while the scan speed was set at 1000 $m/z$. The mobile phase was composed of aqueous ammonium formate (5 mmol/L) with 0.1% formic acid (solvent A) and acetonitrile with ammonium formate (5 mmol/L) with 0.1% formic acid (solvent B). The compounds were separated with the following linearly programmed solvent gradient: 0 min (10% B), 10 min (95% B), 2 min (95% B), then equilibrating back to 10% B for 3 min. The flow rate for the column was set at 0.25 mL/min, while the column temperature was set at 40 °C and the injection volume at 10 µL [55].

### 2.8. Statistical Analysis

The plant growth-promoting activity and Hydrolytic enzyme experiments were repeated twice, and statistical analysis was performed as a comparison of means, and *p* values were calculated with *t*-test analysis at 95% confidence level using IMB SPSS Statistics version 26.0 and phylogenetic tree analysis in NCBI database using MEGA program version 7.

## 3. Results

### 3.1. Identification of Strawberry Anthracnose Fungal Pathogen

Isolation and Identification of *Colletotrichum* sp.

The fungal isolate CA0110 produced unicellular, cylindrical conidia with a mean length of 7.25 µm and a mean width of 2.5 µm (mean of 100 measures). Pathogenicity testing confirmed that isolate CA0110 was pathogenic to strawberry fruits and leaves and induced anthracnose symptoms. Anthracnose symptoms typically appear as dark, sunken lesions at the inoculation area on fruits and leaves (Figure 1). The result of molecular identification showed that the sequence similarity of the isolated fungi to the BLAST result was 100% for *C. acutatum* strain 2002ER and *C. acutatum* strain Ca89; the isolate sequence numbers in GenBank are LN826025.1 and MH887549.1 for the ITS gene and GAPDH gene, respectively.

### 3.2. Characteristic Identification of Streptomyces albus CINv1

3.2.1. Morphology of Streptomyces CINv1

The characteristics of the colony of *Streptomyces* CINv1 were observed after 7 days of inoculation. At the maturity stage, we observed the massive production of a yellowish-to-white spore chain forming on the aerial mycelium and also observed the basal mycelium (pale yellow color) under the medium surface. The size of a single colony was around 8 mm. The arrangement of the spore chains was spiraliform, with 10–20 rod-shaped spores per chain (Figure 2).

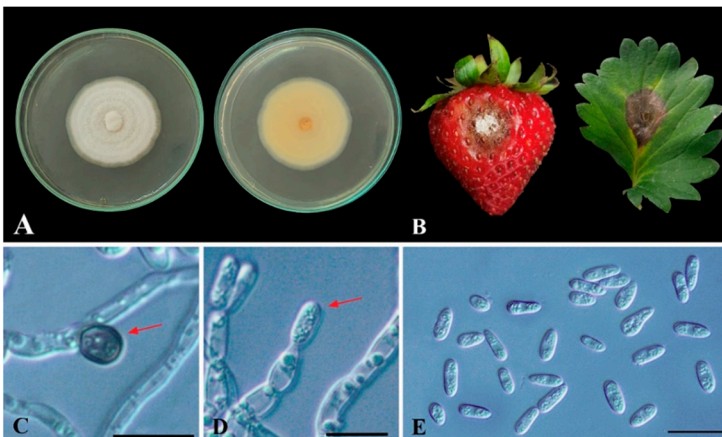

**Figure 1.** *Colletotrichum* sp. isolates CA0110, (**A**): colony on PDA medium, (**B**): pathogenicity test on fruit and leaf, (**C**): appressorium (red arrow), (**D**): conidiogenous cell (red arrow), (**E**): conidia. Scale bars: (**C**–**E**) = 10 μm.

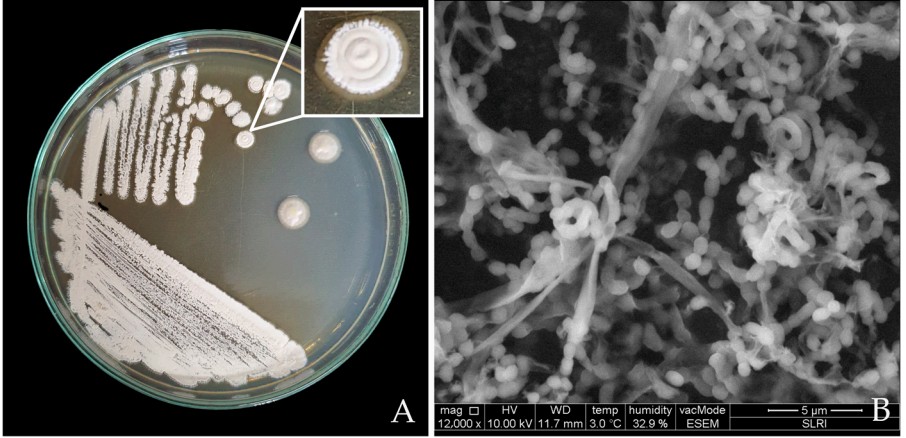

**Figure 2.** (**A**) Colonies of the *Streptomycetes* CINv1 developed on IMA-2 medium after incubation at 30 °C for 7 days, (**B**) Spore chain morphology observed under the Scanning Electron Microscope (SEM, FEI-Quanta 450, Netherlands) at 12,000×.

### 3.2.2. Identification and Phylogenetic Tree of Streptomyces CINv1 Strain

The phylogenetic tree based on 16S rRNA gene sequences showing the relations between *Streptomyces albus* CINv1 and ex-type isolates of diverse species of the genus *Streptomyces* is shown in Figure 3. The neighbor-joining Phylogenetic tree of *Streptomyces* CINv1 shows relationships between CINv1 and related species of the genus *Streptomyces*. This strain clustered with the *S. albus* strains NBRC 13015T, NRRL B-2365T, NBRC 15415T, LMG 20295T, NBRC 13041T, and PRE5, with which it showed 100% similarity. These six strains of *S. albus* were distinct from strain DSM43032T of *S. sclerotiatus*. The 16S rRNA gene sequence of the CINv1 strain was submitted to GenBank and was assigned GenBank accession numbers OR188085.

### 3.2.3. Antifungal Activity of *Streptomyces albus* CINv1 on the Radial Growth of *Colletotrichum* sp. CA0110

*Streptomyces albus* CINv1 was previously reported to effectively inhibit several phytopathogenic fungi. In this study, we investigated the CINv1 strain using the dual culture method. The result showed 81.83% inhibition of the mycelial growth of *Colletotrichum* sp. isolate CA0110 compared with the control treatment, sterile distilled water. During the experiment, the tested fungal colony showed abnormal fungal mycelium. Thus, the edges of the inhibited fungal colonies cultured with the *Streptomyces* isolate were cut,

and the mycelium appearance was observed under the microscope. Abnormal hyphae of *Colletotrichum* sp. isolate CA0110 were observed, including distortion, swelling, and cytoplasmic aggregation, as shown in Figure 4. Meanwhile, the hyphae in the control treatment showed regular radial growth without swelling symptoms.

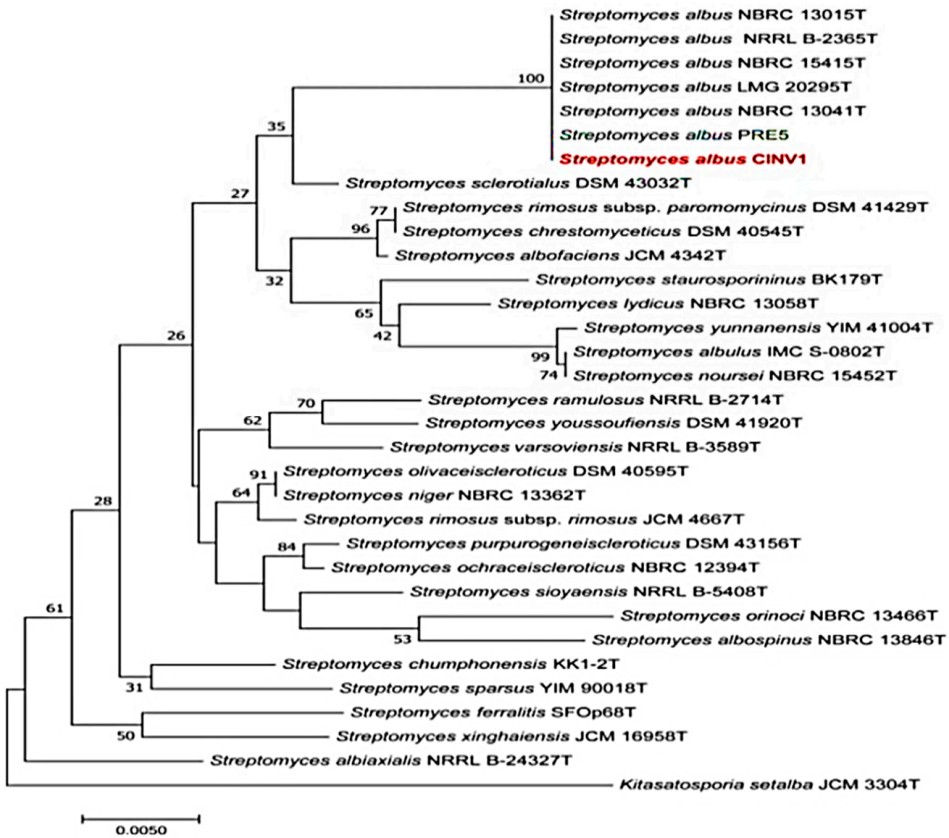

**Figure 3.** Neighbor-joining Phylogenetic tree based on 16S rRNA gene sequences of *Streptomyces* CINv1. Shows relationships between CINv1 and related species of the genus *Streptomyces*. *Kitasatosporia setalba* JCM 3304T was used as outgroup. Numbers at nodes indicate the levels of bootstrap support (%) based on neighbor-joining analysis of 1000 resampled datasets.

### 3.3. Plant Growth-Promotion Activities

The results of tests aimed at evaluating the plant-growth-promoting activities of *S. albus* CINv1 include siderophore production, nitrogen fixation, phosphate, and potassium solubilization, hydrolytic enzyme production, and indole-3-acetic acid (IAA) production (Tables 1 and 2).

**Table 1.** Nitrogen fixation, siderophore production, phosphate solubilizing, potassium solubilizing, and IAA production by *Streptomyces albus* CINv1.

| Treatment | Clear Zone (cm) | | | | IAA Production (µL/100 mL) |
| --- | --- | --- | --- | --- | --- |
| | Phosphate Solubilizing | Potassium Solubilizing | Nitrogen Fixation | Siderophore Production | |
| *Streptomyces albus* CINv1 | 0.38 * ± 0.08 | 0.00 ± 0.00 | + | 0.52 * ± 0.06 | 31.6 ± 0.42 |
| Control | 0.00 ± 0.00 | 0.00 ± 0.00 | − | 0.00 ± 0.00 | − |

* Significant difference ($p < 0.05$), + = positive, − = negative.

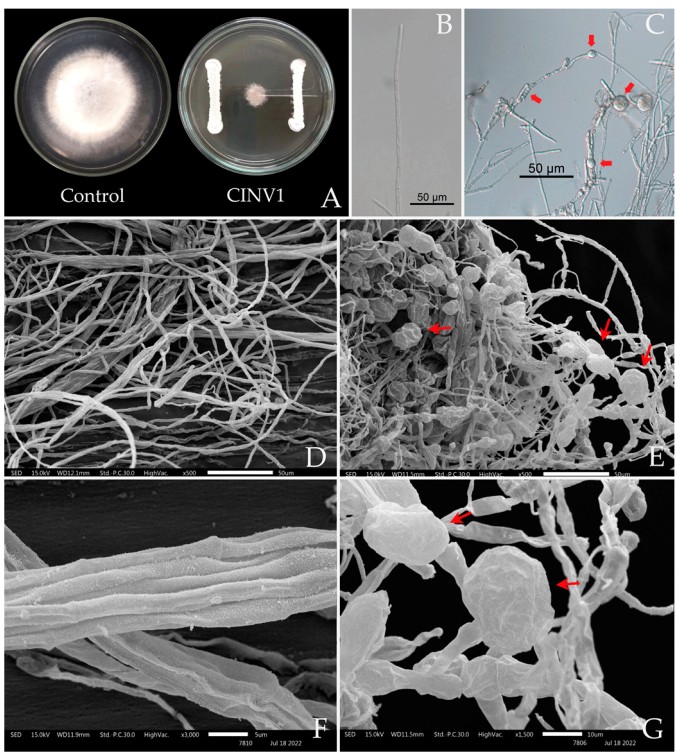

**Figure 4.** Dual culture (**A**) and light microscope images showing hyphal morphologies of *Colletotrichum* sp. CA0110 with and without *Streptomyces albus* CINv1 (**B**,**C**), and Scanning Electron Microscope (SEM) images (**D–G**), showing hyphal morphologies of *Colletotrichum* sp. CA0110 without *Streptomyces albus* CINv1; control (**D**,**F**) and hyphal morphologies of *Colletotrichum* sp. CA0110 with *Streptomyces albus* CINv1 (**E**,**G**), red arrows indicate abnormal hyphal morphologies; Scanning Electron Microscope (SEM) scale bar: 50 (**D**,**E**) and 5 μm (**F**,**G**), respectively.

**Table 2.** Hydrolytic Enzyme Production: protease activity, cellulase assay, chitinase assay, and amylase assay of *Streptomyces albus* CINv1.

| Treatment | Mean Diameter of Halo (cm) | | | |
|---|---|---|---|---|
| | Protease | Cellulase | Chitinase | Amylase |
| *Streptomyces albus* CINv1 | 1.07 * ± 0.08 | 0.76 * ± 0.08 | 0.32 * ± 0.006 | 0.55 * ± 0.03 |
| Control | 0.00 ± 0.00 | 0.00 ± 0.00 | 0.00 ± 0.00 | 0.00 ± 0.00 |

* Significant difference ($p < 0.05$), mean diameter of halo from 15 dishes.

### 3.3.1. Phosphate and Potassium Solubilization Assay

*Streptomyces albus* CINv1 was tested for its phosphate solubilization capacity. The result showed that it induced a clear zone of 0.38 cm in Pikovskaya's medium without bromothymol blue, but a clear zone was not developed in Aleksandrov's modified medium (Table 1).

### 3.3.2. Nitrogen Fixation

Nitrogen fixation was screened by observing the growth in Burk's medium 7 days after incubation in dark conditions. After incubation, *S. albus* CINv1 was able to grow on the plate without utilizing the nitrogen source, which indicates the bacterium's ability to fix nitrogen (Table 1).

### 3.3.3. Siderophore Production

Siderophore production was observed in the color change of the medium from blue to orange on CAS agar plates. The strain showed a clear zone of 0.52 cm at 7 days after incubation in dark conditions (Table 1).

### 3.3.4. Indole-3-Acetic Acid (IAA) Determination

The *S. albus* CINv1 strain has the ability to produce IAA in ISP-2 broth supplemented with L-Tryptophan at a concentration of 31.6 μL/100 mL (Table 1).

### 3.4. Hydrolytic Enzyme Production

Protease activity was observed by the formation of halos in the medium containing skim milk; it developed a clear zone of 1.07 cm. Cellulase activity was observed on the 0.5% CMC agar plate, with the zone of hydrolysis being around 0.76 cm. Meanwhile, chitinase activity was detected by observing the formation of halos around *S. albus* CINv1 in the medium with colloidal chitin. The bacterium developed a clear zone of 0.32 cm. Amylase activity was detected by observing the formation of halos around the *S. albus* CINv1 colony in the medium. It developed a clear zone of 0.55 cm (Table 2).

### 3.5. Plant Growth Promotion Activities

### 3.5.1. Plant Growth Promotion Activities in Tissue Culture Conditions

The results showed that *S. albus* CINv1 had the ability to induce an increase in the number of leaves and roots (9.50 and 10.85, respectively) compared with the control treatment (6.50 and 7.00, respectively). In addition, the bacterium increased canopy height, root length, and total fresh weight (5.27 and 6.81 cm and 0.66 g, respectively) compared with the control treatment (4.93 and 5.95 cm and 0.44 g, respectively). The treatment did not significantly affect the number of stems (Table 3).

**Table 3.** Plant growth promoting *Streptomyces albus* CINv1 in tissue culture conditions after 30 days inoculation.

| Treatment | Plant Growth Parameters | | | | | |
|---|---|---|---|---|---|---|
| | Number of Leaves | Number of Roots | Number of Stems | Canopy Height (cm) | Roots Length (cm) | Total Fresh Weight (g) |
| *S. albus* CINv1 | 9.50 * ± 1.100 | 10.85 * ± 1.694 | 1.05 [ns] ± 0.224 | 5.27 * ± 0.529 | 6.81 * ± 0.786 | 0.66 * ± 0.181 |
| ddH$_2$O | 6.50 ± 1.051 | 7.00 ± 1.974 | 1.00 ± 0.000 | 4.93 ± 0.464 | 5.95 ± 0.573 | 0.44 ± 0.081 |

* Significant difference ($p < 0.05$), ns = not significant, N = 30, mean ± standard deviation.

### 3.5.2. Plant Growth Promotion and Biological Control in Greenhouse Conditions

The results showed that *S. albus* CINv1 could significantly increase the number of leaves (8.38) compared with the control treatment (7.13). In addition, the bacterium significantly increased canopy height (20.60 cm) compared with the control treatment (17.28 cm). The treatment did not significantly affect the number of runners. The biological control of disease in greenhouse conditions shows the result of the percentage of disease incidence at 62.50% (CINv1 treatment), but not significantly when compared with the control treatment at 100.00%. The disease severity showed a score of *S. albus* CINv1 treated significantly at 0.88 (score 1 = very slight), while control treatment showed a disease score of 2.63 (score 3 = severe and dead plant) (Figure 5).

### 3.6. Colonization Study of Streptomyces albus CINv1

The colonization study of the strain in different parts of strawberry plants was studied using SEM. In Figure 6A, scanning electron micrographs (SEM) show that the spore and mycelium penetrated through the stomatal cavity. In addition, the sub-mycelium of *S. albus* CINv1 was attached and penetrated directly into the petiole cell (Figure 6B). Similarly, in

the roots, the mycelium was attached to the epidermis and penetrated directly through the cortex cell, as seen in Figure 6C.

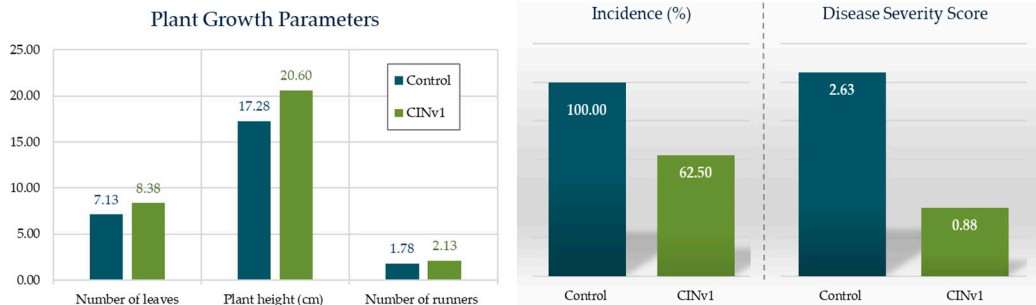

**Figure 5.** Plant growth promotion, % incidence, and disease severity score of *Streptomyces albus* CINv1 compared with control treatment in greenhouse conditions.

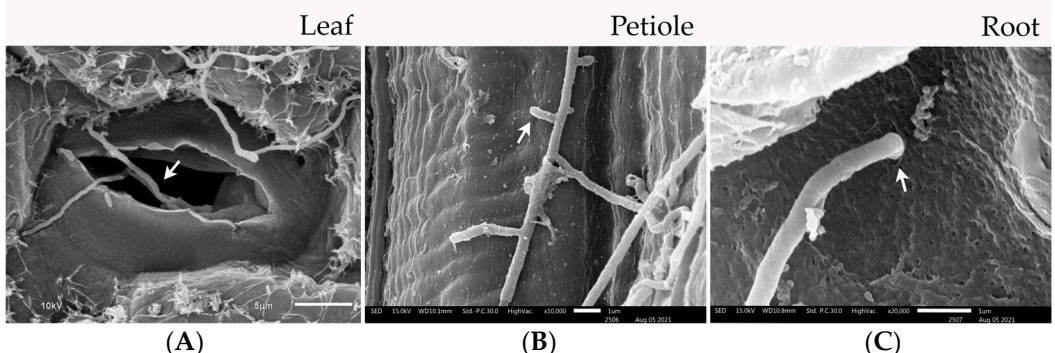

**Figure 6.** Scanning electron micrograph of *Streptomyces albus* CINv1. Images of sub-mycelium (white arrow; sub-myl) colonizing the leaf at the stomata level (**A**), petiole (**B**), and root (**C**) surfaces of strawberry plantlets after 5 days of inoculation in $10^6$ cfµ/mL spore suspension. Scale bars: A = 5 µm, B = 2 µm, C = 1 µm, respectively.

Transmission electron microscopy (TEM) was used to observe the colonization of the root, petiole, and leaf cells of the strawberry plantlets by *S. albus* CINv1. The colonies of this strain were detected as gray and dark particles both inter- and intracellularly. In Figure 7, TEM micrographs show single spores and groups of hyphae of *S. albus* CINv1 located in both intercellular and intracellular cell spaces of the cell wall and cell membrane of the plant. On the leaf samples, TEM micrographs showed spores of *S. albus* CINv1 in the intracellular area of leaf cells (Figure 7A), and we found single spores in the intercellular cell space (Figure 7B). Meanwhile, the sub-mycelium was observed in both the inter- and intracellular spaces (Figure 7A,B). Part of the petiole, spores, and sub-mycelium were detected near the cell membrane in the intracellular zone (Figure 7C,D), but we did not detect any *S. albus* CINv1 cells in the intercellular zone. In the root samples (Figure 7E,F), the sub-mycelium and spores of these strains were found both inter- and intracellularly (Figure 7).

### 3.7. Identification of Streptomyces albus CINv1 Metabolites

Liquid Chromatography/Tandem Mass Spectrometry (LC-MS/MS) was used to analyze the secondary metabolites produced by *S. albus* CINv1. An example of secondary metabolites detected in a liquid medium is listed in Table 4. The compounds contained in the crude list of *S. albus* CINv1 culture filtrate can be separated into groups of antifungal substances, fungicides, and plant growth regulators such as 2,4,6-Tribromophenol and Simeconazole (fungicides), Wybutoxine (fungicides and antifungals), and Tetranactin (anti-

fungals), including plant growth promoters such as Hypotaurocyamine, Phytosulfokine b, and Fusicocin H.

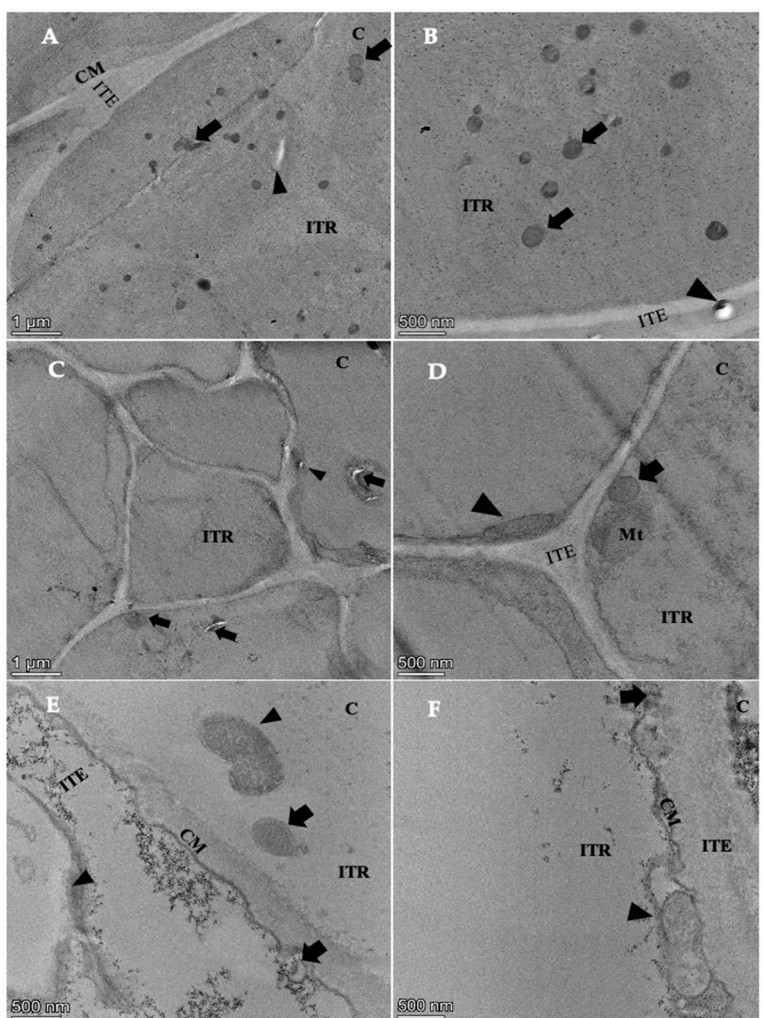

**Figure 7.** TEM images showed colonization of *Streptomyces albus* CINv1 into intercellular and intracellular cells of strawberry plantlet cells. (**A**,**B**): leaf cell; (**C**,**D**): petiole cell; and (**E**,**F**): root cells; ITE: intercellular space; ITR: intracellular space; C: cortex cell; CM: cell membrane. Arrows and arrowheads indicate Sub-myl: Sub-mycelium and C: conidia of CINv1, Mt: mitochondrial, respectively.

**Table 4.** Secondary metabolites from the crude extract of *Streptomyces albus* CINv1, including plant growth promoters as well as plant hormones as determined by LC-MS/MS analysis.

| Compound | Formula | RT * | Mass | *m/z* | Ion Species |
|---|---|---|---|---|---|
| 2,4,6-Tribromophenol | $C_6H_3Br_3O$ | 0.956 | 327.7739 | 372.7722 | (M+HCOO)- |
| Simeconazole | $C_{14}H_{20}FN_3OSi$ | 3.863 | 293.1356 | 292.1283 | (M-H)- |
| Wybutoxine | $C_{16}H_{20}N_6O_7$ | 1.159 | 408.1397 | 453.1379 | (M+HCOO)- |
| Tetranactin | $C_{44}H_{72}O_{12}$ | 3.428 | 792.4996 | 791.4915 | (M-H)- |
| Thiolactomycin | $C_{11}H_{14}O_2S$ | 3.223 | 210.0722 | 269.0862 | (M+CH3COO)- |
| Tolfenpyrad | $C_{21}H_{22}Cl\,N_3O_2$ | 4.722 | 383.1419 | 382.1327 | (M-H)- |
| Hypotaurocyamine | $C_3H_9N_3O_2S$ | 1.018 | 151.0413 | 347.0811 | (2M+HCOO)- |
| Phytosulfokine b | $C_{28}H_{38}N_4O_{14}S_2$ | 1.139 | 718.1708 | 359.0744 | (M-2H)-2 |
| Fusicoccin H | $C_{26}H_{42}O_8$ | 4.116 | 406.1252 | 380.9792 | (M-H)- |

* RT = Retention Time. *m/z* = Counts vs. Mass-to-Charge (*m/z*).

## 4. Discussion

Several bacterial strains, including *Bacillus* spp., *Pseudomonas* spp., *Serratia* spp., *Paenibacillus* spp., *Azotobacter* spp., and *Streptomyces* spp., have been tested as biological control agents (BCAs) and plant growth-promoting rhizobacteria (PGPR) [56]. Most studies, including ours, have focused on the antagonistic action of these bacteria, which involves suppressing pathogen growth by producing antifungal compounds [57,58]. In this study, the potential of the CINv1 strain was evaluated based on its abilities to be a biological control agent, a plant growth promoter, and an endophyte. Thus, initially, we have undertaken the task of identifying the fungal pathogen using morphological and molecular identification and assessing its pathogenicity. The results showed *Cacutatum*, and our findings agree with previous studies [2]. Moreover, identification through molecular techniques is one of the important criteria that facilitates the categorization of organisms at the species level. *Colletotrichum* spp. is a species complex with phylogenetic analyses involving multiple gene sequences [8].

Previous studies have also explored the use of beneficial fungi, bacteria, and endophytic *Streptomyces* for the biological control of strawberry anthracnose [2,4] and anthracnose disease in fruits [59]. However, such approaches are relatively rare in Thailand. A phylogenetic analysis of *Streptomyces* based on 16S rRNA sequences confirmed the species. In our study, the 16S rRNA sequences analyzed indicated 100% identity for *Streptomyces albus*. Moreover, *S. albus* species have various bioactive compounds. Such as when *S. albus* POR-04-15-053 was isolated from a marine sediment, it was effective for antitumor production [60], *S. albus* Tia1 was effective for the biochemical nature of the lytic principal production [61], and *S. albus* CAI-21 isolate from herbal vermicompost was effective for antifungal compound production [62]. Thus, the host of *Streptomyces* can indicate a group of secondary metabolites and their application. In our study, the CINv1 strain produced various effective antifungal compounds and plant growth-promoting substances and showed mechanisms to colonize host cells. The effectiveness of the CINv1 strain against the strawberry pathogen *Colletotrichum* sp. CA0110 was clearly proved by demonstrating significant inhibiting results on the pathogen growths, including abnormal swelling and bulbous structures of hyphae [63]. Similarly, *B. amyloliquefaciens* strain Bam22 inhibited the mycelium growth of *S. sclerotiorum* and turned it into "balloons" [58,59,63]. (Figure 4). Additionally, LC-MS/MS techniques were used to identify bioactive compounds. In our study, a secondary metabolite from CINv1 strain culture filtrate was determined by a non-targeted LC-MS/MS approach coupled with spectral networking. These techniques were appropriate for quick screening of secondary metabolites [64]. The screening results showed various groups of bioactive compounds such as fungicides (2,4,6-Tribromophenol, Simeconazole, and Wybutoxine) and antifungals (Tetranactin, Thiolactomycin, and Tolfenpyrad) and plant hormones (Hypotaurocyamine, Fusicoccin H, and Phytosulfokine b) (Table 4) that indicated the effectiveness of the strain.

In agriculture, microorganisms that support plant growth and disease control have become effective substitutes for synthetic pesticides [65]. Numerous fungal and bacterial phytopathogens can potentially be controlled by antagonistic yeast, fungal endophytes, bacteria, actinobacteria, and their metabolites [66–68]. Similar results were reported by [68], in our study, the CINv1 strain exhibited both direct and indirect mechanisms of plant growth promotion. The direct mechanism involved nitrogen fixation, phosphate solubilization, siderophore production, and the synthesis of phytohormones such as IAA, hypotaurocyamine, Phytosulfokine b, and fusicoccin H. The indirect mechanism included the production of hydrolytic enzymes and antibiotics to degrade fungal cell walls and enhance immune responses. However, the results of plant growth promotion activity on medium agar plates and plant hormone production showed a relationship with tissue culture testing and greenhouse conditions trials, which confirmed the efficacy of *S. albus* CINv1 for promoting strawberry growth.

A large number of actinobacteria interact closely with eukaryotes like fungi, insects, animals, and plants. The colonization study of bacterial endophytes in the plant cell

indicated mutualistic relationships. In our study, the CINv1 strain colonized plant cells and enhanced plant health. SEM and TEM analysis showed the ability of CINv1 spores and mycelium to enter strawberry cells, aligning with the same previous reports [25,69]. Similar colonization patterns have been observed in other endophytic bacteria, such as *B. subtilis* and *Paenibacillus* spp. These bacteria establish a sustainable relationship within plant cells, promoting plant growth and health [70].

## 5. Conclusions

Strawberry production is the most popular horticultural crop in northern Thailand. However, due to the high disease susceptibility, high amounts of fungicides are being used. Hence, the use of antagonistic microbes is promoted under sustainable and green agriculture concepts. In our study, the fungal pathogen and *Streptomyces* CINv1. were identified, and the results showed *Colletotrichum* sp. CA0110 and *S. albus* CINv1, respectively. The CINv1 strain has the potential to control the fungal pathogen in laboratory and greenhouse conditions. In addition, this CINv1 strain can produce plant hormones, Siderophore production, and hydrolytic enzyme production. Furthermore, *S. albus* CINv1 has the ability to colonize both inter- and intracellularly in the leaf, root, and petiole cells of strawberry plantlets. The results were further confirmed through field studies.

**Author Contributions:** Conceptualization, K.K.; methodology, K.K.; validation, W.P., O.-U.R. and K.K.; formal analysis, W.P. and N.T.; investigation, O.-U.R. and K.K.; resources, K.K.; writing—original draft preparation, W.P.; writing—review and editing, W.P.; visualization, K.K.; supervision, K.K.; project administration, W.P. and K.K.; funding acquisition W.P. and K.K. All authors have read and agreed to the published version of the manuscript.

**Funding:** This research was funded by the Agricultural Research Development Agency (Public), ARDA, 2003/61, Phaholyothin Road, Lat Yao, Chatuchak, Bangkok, 10900.

**Data Availability Statement:** Not applicable.

**Acknowledgments:** The authors would like to thank the staff of the laboratories and greenhouse at the Faculty of Agriculture, Chiang Mai University, for their technical supports.

**Conflicts of Interest:** The authors declare no conflict of interest.

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
