# Peer review of "Plant Growth Promoting and Colonization of Endophytic Streptomyces albus CINv1 against Strawberry Anthracnose"

_horticulturae, doi:10.3390/horticulturae9070766_

Round 1
Reviewer 1 Report
COMMENTS TO THE MANUSCRIPT “Plant Growth Promoting and Colonization of Endophytic Streptomyces albus CINv1 Against Strawberry Anthracnose” by Pupakdeepan et al.
General comment:
The submitted manuscript describes the isolation of one Streptomyces and one Colletotrichum strains, its morphological and molecular characterization and the antagonism and biocontrol of the former against the latter. Also, the plant growth promotion and the endophytic plant colonization of the Streptomyces strain is analyzed.
In general, the submitted manuscript is well written, the methodology is properly described, and the results clearly explained. The Discussion section can be improved.
The subject of the submitted manuscript is suitable to be published in the Horticulturae journal. However, the manuscript needs a major review and optimization before being accepted. In the specific comments below, I detail the bases for my decision.
Specific comments:
1. I suggest that the aim of the submitted work must be explicit mentioned at the end of the Introduction section.
2. Use h as acronym of hours, not hr (line 134).
3. I suggest locating subsection 2.3. Characteristics Identification of Endophytic Streptomyces CINv1 after the subsection 2.1. Confirmation of Strawberry Anthracnose Fungal Pathogen. In this way you describe first the isolation and identification of both the pathogen and the biocontrol bacteria. The subsections 2.1.2. and 2.1.3 can be included in a 2.2 subsection to describe the methods to analyze the interactions between both microorganisms. If this re-location is made, then the rest of the Materials and Methods subsections must be renumbered. I suggest the same re-location in the Results section, putting first the results of the Streptomyces strain identification and then those of the interaction with the fungal pathogen.
4. Did you conduct the Blasn search of your 16S sequence in the curated database of this gene for bacteria? Please provide a detailed description of how you conducted the phylogenetic analysis of the Streptomyces isolate: how was the multiple sequence alignment conducted?, How was the evolutive model selected and what it was?, What was the criteria used to generate the phylogenetic tree? How many bootstrap iterations were used?
5. In the same way that for bacteria, did you conduct the ITS sequence identity search in the Blastn ITS curated fungal sequences or in the general Blastn? Also, if you generated the ITS and GAPDH PCR products for your Colletotrichum strain, why do not conduct a phylogenetic analysis. It is better to show a phylogenetic tree than the simple result of the Blastn search. This is even more relevant if you did not conduct the Blastn search in the ITS fungal curated database.
6. Please specify what the ND acronym means in Table 2.
7. Please provide additional information on the metabolites synthetized by Streptomyces (Adduct and Main fragment ions m/z for each). These are relevant data to ensure correct identification.
8. Please correct the name streptomyces in the figure captions 7 and 8, by writing in capital the first letter. The same for the name in lines 469 and 476.
9. Please write in italics Streptomyces in the lines 474 and 495.
10. The Discussion section is very general and can be improved by contrasting the obtained results of the submitted manuscript with some of the numerous previous studies that characterizes isolates of Streptomyces spp. and its interaction with Colletotrichum spp. Also, please contrast the plant growth promotion efficiency of your Streptomyces strain with previous S. albus studies in this regard.
Author Response
Response to Reviewer 1 Comments
Point 1: I suggest that the aim of the submitted work must be explicit mentioned at the end of the introduction section.
Response 1: Thank you for your kind suggestion. we have added the aim at the end of the introduction (line 95–101).
Point 2: Use h as acronym of hours, not hr (line 134)
Response 2: we are corrected “h” in line 134
Point 3: I suggest locating subsection 2.3. Charecteristics Identification of Endophytic Streptomyces CINv1 after the subsection 2.1. Confirmation of Strawberry Anthracnose Fungal Pathogen. In this way you describe first the isolation and identification of both the pathogen and the biocontrol bacteria. The subsections 2.1.2. and 2.1.3. can be included in a 2.2 subsection to describe the methods to analyze the interaction between both microorganisms. If this re-location is made, then the rest of the Materials and Methods subsections must be renumbered. I suggest the same re-location in the Results section, putting first the results of the Streptomyces strain identification and then those of the interaction with the fungal pathogen.
Response 3: Following your suggestion, we improved the re-location is made in subsections 2.2-2.3. (line 126-192) and 3.2-3.3 (line 334-377)
Point 4: Did you conduct the Blast search of your 16S sequence in the curated database of this gene for bacteria? Please provide a detailed description of how you conducted the phylogenetic analysis of the Streptomyces isolate: how was the multiple sequence alignment conducted? How was the evolutive model selected and what it was?, What was the criteria used to generate the phylogenetic tree? How many bootstrap iterations were used?
Response 4: Following your suggestion, we improved the methodology (line 161-170).
Point 5: In the same way that for bacteria, did you conduct the ITS sequence identity search in the Blastn ITS curated fungal sequences or in the general Blastn? Also, if you generated the ITS and GAPDH PCR products for your Colletotrichum strain, why do not conduct a phylogenetic analysis. It is better to show a phylogenetic tree than the simple result of the Blastn search.
Response 5: Thank you for your kind suggestion. This paper aims to demonstrate the efficacy of Endophytic S. albus CINv1 in controlling Strawberry Anthracnose. It is noteworthy that the Colletotrichum acutatum species, known for its high degree of complexity, requires the utilization of multiple genes for accurate species-level classification, key genes such as ITS, GAPDH, CHS-1, ACT, and TUB2. However, in this study, the focus is on showcasing the control capabilities of the aforementioned endophytic strain, rather than identifying the specific pathogen species causing Strawberry Anthracnose.
Point 6: Please specify what the ND acronym means in Table 2.
Response 6: ND: Not detected, but we will be changed “ - ” (after relocate shown in Table 1 ).
Point 7: Please provide additional information on the metabolites synthetized by Streptomyces (Adduct and Main fragment ions m/z for each). These are relevant data to ensure correct identification.
Response 7: This detail has been added in subsections 3.7, Table 4. Columns 5-6.
Point 8: Please correct the name streptomyces in the figure captions 7 and 8 (6 and 7), by writing in capital the first letter. The same for the name in lines 469 and 476.
Response 8: I have corrected the name streptomyces in the figure captions 6 in line 446 and 7 in line 464,
Point 9: Please write in italics Streptomyces in the lines 474 and 495.
Response 9: I have corrected the italics Streptomyces in all the manuscript
Point 10: The discussion section is very general and can be improved by contrasting the obtained results of the submitted manuscript with some of the numerous previous studies that characterize isolates of Streptomyces spp. and its interaction with Colletotrichum spp. Also, please contrast the plant growth promotion efficiency of your Streptomyces strain with previous S. albus studies in this regard.
Response 10: Following your suggestion, we improved the discussion section.
Reviewer 2 Report
1. The abstract should show a logical background and need of the study, then there is the objective. Moreover, the source of materials, as the bacteria, the strawberry, and the pathogen should be briefly included in this section. Furthermore, this abstract does not show methods. However, there are unnecessary details in the abstract, such as those discussing about mechanisms and listing substances, which are not revealed by statistical data. Such data should appear. Ultimately, the current version of the abstract is wordy, but incomplete. It did not illustrate a proper and logical flow of works, though it does indicate what the study can offer. Thus, the current abstract is inappropriate, and needs to be carefully revised.
2. Vocabulary, spelling and grammar mistakes are found in the manuscript. They should be corrected in the final version.
3. Lines 67-70 and 87-90: If possible, please find some literatures to cite these statements.
4. Although the introduction informs sufficiently, it did not have neither the need why the study should be conducted, nor the objectives or hypothesis of the current study.
5. Names of the species are not correctly formatted in the subheadings and text. This should be noticed and more careful.
6. The source of the Streptomyces CINv1 is should be before the section 2.1.2.
7. The number format should be uniform. Some appear with a coma, some are not, for numbers above 999.
8. Some terms have been abbreviated, but still appear in full forms.
9. Lines 320-322: this sentence is ambiguous and confused. It can lead to misunderstanding. Therefore, please revise it.
10. Some information in Table 1 is derived from other studies which should be cited. However, citations should not be allowed in the results. I suggest moving the table to the discussion.
11. The Streptomyces CINv1 had not been identified yet, but the name Streptomyces albus CINv1 appeared from the 3.1.2 section. This is illogical. Moreover, after first advent as Streptomyces albus, S. albus should be used.
12. Tables should appear when they are first mentioned. This should be checked.
13. This “Transmission electron microscopy (TEM)” should appear first in the materials and methods.
14. Lines 500-507: This paragraph is not necessary for the discussion. This version of the discussion is poor. It should be improved and expanded.
15. The conclusion contains unnecessary items. The conclusion should summarize the whole result with data, and provide the future prospect of the study. It is not responsible for explanation and discussion.
Minor editing of English language required
Author Response
Response to Reviewer 2 Comments
Point 1: The abstract should show a logical background and need of the study, then there is the objective. Moreover, the source of materials, as the bacteria, the strawberry, and the pathogen should be briefly included in this section. Furthermore, this abstract does not show methods. However, there are unnecessary details in the abstract, such as those discussing about mechanisms and listing substances, which are not revealed by statistical data. Such data should appear. Ultimately, the current version of the abstract is wordy, but incomplete. It did not illustrate a proper and logical flow of works, though it does indicate what the study can offer. Thus, the current abstract is inappropriate, and needs to be carefully revised.
Response 1: Following your suggestion, we improved the abstract in lines 11-30
Point 2: Vocabulary, spelling, and grammar mistakes are found in the manuscript. They should be corrected in the final version.
Response 2: Following your suggestion, we corrected the vocabulary, spelling, and grammar in the manuscript.
Point 3: Lines 67–70 and 87–90: If possible, please find some literatures to cite these statements.
Response 3: Thank you for your kind suggestion, we improved the final version.
Point 4: Although the introduction informs sufficiently, it did not have neither the need why the study should be conducted, nor the objectives or hypothesis of the current study.
Response 4: Thank you for your kind suggestion. we have added the aim at the end of the introduction in lines 95–101.
Point 5: Names of the species are not correctly formatted in the subheadings and text. This should be noticed and more careful.
Response 5: Following your suggestion, we corrected the name of the species.
Point 6: The source of the Streptomyces CINv1 is should be before the section 2.1.2.
Response 6: Following your suggestion, we relocating the source of the Streptomyces CINv1 before section 2.1.2. in line 126-135.
Point 7: The number format should be uniform. Some appear with a comma, some are not, for numbers above 999.
Response 7: Following your suggestion, corrected the number format.
Point 8: Some terms have been abbreviated, but still appear in full forms.
Response 8: Following your suggestion, corrected the abbreviated.
Point 9: Lines 320–322: this sentence is ambiguous and confused. It can lead to misunderstanding. Therefore, please revise it.
Response 9: In this paragraph, we have revised in line 362-364.
Point 10: Some information in Table 1 is derived from other studies which should be cited. However, citations, should not be allowed in the results. I suggest moving the table to the discussion.
Response 10: Following your suggestion, we deleted the citation data (the last column) and moved to the discussion section.
Point 11: The Streptomyces CINv1 had not been identified yet, but the name Streptomyces albus CINv1 appeared from the 3.1.2 section. This is illogical. Moreover, after first advent as Streptomyces albus, S. albus should be used.
Response 11: Following your suggestion, we have corrected the name Streptomyces albus CINv1 in 3.1.2 and edited the word S. albus after the first advent.
Point 12: Tables should appear when they are first mentioned. This should be checked.
Response 12: Following your suggestion, we checked and edited all the Tables.
Point 13: This “Transmission electron microscopy (TEM)” should appear first in the materials and methods.
Response 13: Thank you for your kind suggestion. Transmission electron microscopy (TEM) is a part of the colonization study. These parts, we would like to present these after scanning electron microscopy (SEM).
Point 14: Line 500–507: This paragraph is not necessary for the discussion. This version of the discussion is poor. It should be improved and expanded.
Response 14: This paragraph has been deleted (Line 500–507) following your suggestion. And we have improved all discussions in line 482-541.
Point 15: the conclusion contains unnecessary items. The conclusion should summarize the whole result with data, and provide the future prospect of the study. It is not responsible for explanation and discussion.
Response 15: In the conclusion section, we have edited your suggestion in line 543-553.

Reviewer 3 Report
A brief summary
The effect of the endophytic Streptomycetes albus, strain CINv1 on the causative agent of strawberry anthracnose Colletotrichum sp. both in vitro in tissue culture and on infected strawberry plants in a greenhouse was investigated. It was shown that strain CINv1 significantly suppresses the development of colonies of Colletotrichum sp. on Petri dishes. The strain CINv1 reduces strawberry disease incidence and disease severity in greenhouse conditions. In addition, strain CINv1 produces plant hormones and secondary metabolites of groups that stimulate plant growth. The endophytic Streptomycetes albus strain CINv1 effectively counteracts strawberry anthracnose and stimulates the growth of the host plant.
Broad comments
The endophytic Streptomyces albus strain CINv1 demonstrated potential as a biological control agent and plant growth-promoting rhizobacteria. The strain CINv1 showed the ability to colonize strawberry cells, contributing to plant health and development. These findings are important for understanding how biological control agents and plant growth-promoting rhizobacteria control plant pathogens and promoting plant growth.
Specific comments
Lines 440, 459, 476: streptomyces change to Streptomyces
Author Response
Response to Reviewer 3 Comments
Point 1: Lines 440, 459, 476: streptomyces change to Streptomyces
Response 1: we have edited the word Streptomyces in line 446, 464, and all the manuscript.
Thank you very much for your valuable suggestion
Submission Date
22 June 2023

Round 2
Reviewer 1 Report
COMMENTS TO THE MANUSCRIPT “Plant Growth Promoting and Colonization of Endophytic Streptomyces albus CINv1 Against Strawberry Anthracnose” by Pupakdeepan et al.
I thank the authors for considering most of the suggestions and comments I made to their manuscript.
I consider that there are still some corrections to do before accepting manuscript for publication. Specifically, the phylogenetic analysis is not clearly explained.
Regarding the phylogenetic tree (Figure 4), please kindly indicate:
i. Did you conduct the Blastn search in the 16S bacterial curated database of GenBank or in the general database? The curated database allows retrieving sequences with the certainty of taxonomic identity, whereas the general database did not.
ii. How was the multiple sequence alignment conducted? (Did you use the Muscle or Clustal algorithm within MEGA?)
iii. How did you select the evolutive model to reconstruct the phylogenetic tree?
iv. Why did you use the Neighbor Joining criterion to reconstruct the phylogenetic tree instead of Maximum Likelihood, which is robust for phylogenetic inferences?
v: How many bootstrap iterations were applied to the phylogenetic tree? (100, 200, 500?)
It is necessary to include all these parameters in any phylogenetic reconstruction.
I asked the same questions in the previous manuscript review, but you do not answer any in the new version of the manuscript.
Finally, it is highly recommendable to submit the 16S sequence of the studied Streptomyces strain to the GenBank database and include the accession number generated in the published manuscript. This is desirable for each new strain analyzed in order to share your data with the scientific community and allow to verify the sequence quality.
Author Response
29th June 2023
Dear Review 1
Our manuscript entitled “Plant Growth Promoting and Colonization of Endophytic Streptomyces albus CINv1 Against Strawberry Anthracnose” (Manuscrip ID: horticulturae-2352387)
We revised your suggestion as the file attaches to the bottom. And, we revised (round 2) the manuscript, the text as the red color was under the tracking in MS, and wrote the response into the response form as attached in the file.
we checked and edited all references and we rephrase similar sentences
Please, consider the opportunity to publish this manuscript. We look forward to hearing from you at your earliest convenience.
Sincerely,
Associate Professor Dr. Kaewalin Kunasakdakul
Department of Entomology and Plant Pathology,
Faculty of Agriculture, Chiang Mai University,
Chiang Mai 50200, Thailand
Phone No.: +66-94941-6223
Email Address: kaewalin.k3@gmail.com
Response to Reviewer 1 Comment Round 2
I asked the same questions in the previous manuscript review, but you do not answer any in the new version of the manuscript.
Response: We sincerely apologize to you. We do not respond to clarification on each issue, because we have only described it in subsection 2.2.2 (line160-169). And we improved the Figure 3 description (line354-358).
Specifically, the phylogenetic analysis is not clearly explained. Regarding the phylogenetic tree (Figure 4), please kindly indicate:
Point 1: Did you conduct the Blastn search in the 16S bacterial curated database of GenBank or in the general database?. The curated database allows retrieving sequences with the certainty of taxonomic identity, whereas the general database did not.
Response 1: Yes. We conduct the Blastn search in the 16S bacterial curated database of GenBank.
Point 2: How was the multiple sequence alignment conducted? (Did you use the Muscle or Clustal algorithm within MEGA?)
Response 2: We used MAFFT version 7 to alignment within MEGA.
Point 3: How did you select the evolutive model to reconstruct the phylogenetic tree?
Response 3: We used the MEGA version 7 program and the base model is Jukes-Cantor model.
Point 4: Why did you use the Neighbor Joining criterion to reconstruct the phylogenetic tree instead of Maximum Likelihood, which is robust for phylogenetic inferences?
Response 4: We used the Neighbor-Joining criterion to reconstruct the phylogenetic tree because we search for information and this strain is not a new species and many reports of Streptomyces albus strain showed these methods to reconstruct the phylogenetic tree.
Point 5: How many bootstrap iterations were applied to the phylogenetic tree? (100, 200, 500?)
It is necessary to include all these parameters in any phylogenetic reconstruction.
Response 5: We used 1,000 bootstrap iterations applied to the phylogenetic tree.
Finally, it is highly recommendable to submit the 16S sequence of the studied Streptomyces strain to the GenBank database and include the accession number generated in the published manuscript. This is desirable for each new strain analyzed in order to share your data with the scientific community and allow to verify the sequence quality.
Response: Following your recommenced, we have submitted the 16S sequence of Streptomyces albus CINv1 to the GenBank database the results showed accession number OR188085. And we added information in line 358-359.
Thank you very much for your valuable suggestion
29 June 2023
